# Length-Adaptive Distillation:
# Customizing Small Language Model for Dynamic Token Pruning

**Chang Liu**[1,2], **Chongyang Tao**[3], **Jianxin Liang**[1,4], **Jiazhan Feng**[1,4], **Tao Shen**[5],
**Quzhe Huang**[1,4], **Dongyan Zhao**[1,2,6,7*]

[1] Wangxuan Institute of Computer Technology, Peking University
[2] Center for Data Science, Peking University
[3] Microsoft Corporation
[4] School of Intelligence Science and Technology, Peking University
[5] AAII, University of Technology Sydney
[6] Institute for Artificial Intelligence, Peking University
[7] State Key Laboratory of Media Convergence Production Technology and Systems

{liuchang97,liangjx,fengjiazhan,huangquzhe,zhaody}@pku.edu.cn
{chotao}@microsoft.com      tao.shen@uts.edu.au

## Abstract

Pre-trained language models greatly improve the performance of various tasks but at a cost of high computation overhead. To facilitate practical applications, there are mainly two lines of research to accelerate model inference: model compression and dynamic computation (e.g., dynamic token pruning). Existing works either adopt these methods individually or simply apply dynamic computation approaches upon a compressed small language model. We argue that they are sub-optimal since the two approaches are separately designed so the compressed model may not be tailored for dynamic computation. To tackle this problem and make compressed small language models faster, we propose Length-Adaptive Distillation, a two-stage knowledge distillation framework that aims to produce a customized small language model for dynamic token pruning. In the general distillation stage, we enforce the student to mimic and reconstruct the teacher's output based on the dynamically pruned representations. Then in the task-specific distillation stage, the student is further accustomed to token pruning while absorbing the task-specific knowledge. Experimental results on GLUE benchmark demonstrate that our method can make the small language model more customized for dynamic token pruning and achieve better speed-performance trade-off.

## 1 Introduction

With the rapid progress of pre-trained language models (PLMs) (Devlin et al., 2019; Liu et al., 2019; Brown et al., 2020), dramatic improvement has been achieved in a wide range of natural language understanding and generation tasks. Despite the remarkable performance those PLMs have

achieved, they generally suffer from high computation overhead, which prevents them from being deployed into resource-scarce applications. Hence, there is an urgent need to accelerate inference speed while minimizing performance degradation.

To achieve this goal, great efforts have been made, among which model compression and dynamic computation are two major lines of research with different mechanisms. Model compression approaches typically convert the large model into a smaller one with reduced layer number and hidden dimension, while the computation graph of the compressed small model is fixed during inference. One of the effective approaches to achieve this goal is knowledge distillation from a large teacher model to a small student model (Jiao et al., 2020a; Wang et al., 2021; Liu et al., 2022a). As for dynamic computation methods, they don't change the architecture of the given language model and dynamically prune its computation graph during inference instead. There are two perspectives in dynamic computation for transformer-based (Vaswani et al., 2017) language models: depth-level early exiting (Liu et al., 2020; Xin et al., 2020; Zhou et al., 2020) and width-level token pruning (Goyal et al., 2020; Kim and Cho, 2021; Kim et al., 2021). We focus on token pruning methods in this paper.

Since these two groups of methods achieve inference acceleration from two orthogonal perspectives, a natural idea is to combine them together to obtain faster inference speed. The simplest way to combine them is the pipeline approach (Kim and Cho, 2021; Guskin et al., 2021, 2022): preparing a compressed small language model first and applying dynamic computation algorithm upon it next. Though proved effective, we argue that the pipeline approach is sub-optimal since model compression

---

* Corresponding author: Dongyan Zhao.

and dynamic computation procedures are individually designed and may not be compatible with each other that much. As a result, the potential of such an approach has not been fully exploited.

To address this issue and achieve better speed-performance trade-off, we propose a novel knowledge distillation framework named Length-Adaptive Distillation (abbr. **LAD**) that transfers the knowledge from the large language model to a small language model while helping the small model to get adapted to dynamic token pruning. Drawing inspirations from Jiao et al. (2020b), we adopt a two-stage distillation paradigm where a general-purpose small language model is first distilled on large-scale general corpora and then task-specific knowledge is injected on task-specific datasets. In general distillation, we mainly consider two issues: (1) how to transfer high-quality general knowledge from the teacher and (2) how to customize the student model for dynamic token pruning. For (1), we propose to transfer token-level knowledge of two types: hidden representations that encode the contextual semantic information of tokens and attention dependency that is generally adopted as the indication of token importance (Goyal et al., 2020; Kim and Cho, 2021) for token pruning. The alignment of hidden representations between the teacher and the student is achieved by contrastive distillation and the matching of attention dependency is fulfilled by mean square error. While for (2), we employ a teacher with an unpruned computation graph to teach a student with dynamic token pruning where the importance of tokens is measured by the attention scores, and the retention configuration is sampled from a uniform distribution. As only the unpruned tokens have their representations in the student's last layer, we train the student by enforcing the remaining token representations to be close to the teacher's corresponding representations as well as to reconstruct the complete token representations of the teacher's. Then in task-specific distillation, we first employ data augmentation to enlarge the task datasets, then distill the student following the dynamic token pruning setting used in general distillation but transfer more task-relevant knowledge by enforcing the sentence embedding of the student to be close to the teacher via contrastive distillation.

We conduct experiments on GLUE benchmark (Wang et al., 2018). We first prove that our knowledge distillation method outperforms advanced knowledge distillation methods on a standard setting (i.e., without token pruning). Moreover, we prove that with the help of the customized length-adaptive distillation, our methods successfully take advantage of both model compression and dynamic computation, and achieve dramatically better speed-performance trade-off compared with the pipeline approach. We release our implementation to facilitate future research[1].

To sum up, our contributions are three folds:

• We propose a two-stage knowledge distillation framework LAD that effectively combines model compression and dynamic computation to achieve faster speedup in inference.

• We customize a small language model for dynamic token pruning by cast a length-adaptive setting on the student and enforcing it to mimic and reconstruct the teacher's representations.

• We conduct comprehensive experiments on the GLUE benchmark and verify that our method can achieve a superior speed-performance trade-off compared with other methods.

## 2 Related Work

### 2.1 Language Model Compression

There are various methods to compress a large language model into a small one including pruning (Fan et al., 2019; Gordon et al., 2020), quantization (Zafrir et al., 2019; Shen et al., 2020; Bai et al., 2021), weight sharing (Dehghani et al., 2018; Lan et al., 2019), knowledge distillation (Hinton et al., 2015; Sanh et al., 2019; Jiao et al., 2020b) and so on. We focus on knowledge distillation in this paper, where a large model acts as the teacher and transfers its knowledge to a smaller student model. Jiao et al. (2020b) proposed general-then-task-specific distillation framework and transferred the knowledge in hidden states, attention matrices, and output logits in different distillation stages. Wang et al. (2020b, 2021) matched the attention dependencies derived from the query, the key and the value vectors in the self-attention module. Sun et al. (2020); Fu et al. (2021) employed contrastive distillation to match the hidden states. Park et al. (2021); Liu et al. (2022a) structured the knowledge as the relations of hidden states and enforced the relations between the teacher and the student to be consistent. Compared with existing works, our method not only addresses the issue of improving the performance of small language models with

---

[1] https://github.com/EMNLP-LAD/LAD

fixed computation graphs as they did, but also endows the small language model with good adaptation ability for dynamic token pruning to achieve improved inference efficiency.

## 2.2 Dynamic Computation

Different from language model compression methods that produce a small model with fixed computation graph, dynamic computation approaches fix the model architecture and achieve inference acceleration by dynamically pruning its computation graph (Han et al., 2021). Liu et al. (2020); Xin et al. (2020); Zhou et al. (2020) studied depth-level early exiting where samples with different difficulty are output from different layers (i.e., the more difficult the sample is, the more layers it would go through before getting its final output). While (Goyal et al., 2020; Kim and Cho, 2021; Kim et al., 2021; Ye et al., 2021; Guan et al., 2022; Modarressi et al., 2022; Liang et al., 2023) proposed width-level token pruning where unimportant tokens are progressively removed as the calculation goes from shallow to deep layers. Among the two lines of research, we focus on token pruning approaches. Recently, Guskin et al. (2021, 2022) explored the combination of dynamic token pruning and model compression to make more aggressive acceleration by simply applying token pruning upon a given compressed language model in a pipeline manner. Different from these works, we highlight the importance the adaptation from a fixed computation graph to a dynamic one for small language models, and propose a knowledge distillation framework to fulfill this goal.

## 3 Methodology

Our framework adopts a two-stage knowledge distillation paradigm (Jiao et al., 2020b) where a randomly initialized small language model is first distilled with general knowledge, and is then taught with task-specific knowledge. Along with the distillation process, the student not only absorbs the knowledge transferred from the teacher, but also gets adapted to dynamic token pruning. After the two distillation stages, we employ evolutionary search (Cai et al., 2019; Wang et al., 2020a; Kim and Cho, 2021) to obtain a set of length configurations for token pruning with different speedups. In the following subsections, we first introduce our knowledge distillation algorithms in general and task-specific distillation stages, then demonstrate

how to apply token pruning upon our model.

## 3.1 General Distillation

General distillation aims to transfer the general knowledge held in a pre-trained language model (i.e., RoBERTa$_{base}$) to a randomly initialized small language model that has fewer layers and hidden dimensions with large-scale open-domain corpora (i.e., Wikipedia). Existing works (Sanh et al., 2019; Jiao et al., 2020b; Sun et al., 2020; Wang et al., 2021) have designed various effective algorithms by considering different knowledge sources, knowledge types, distance metrics, etc. However, they all focus on improving the evaluation performance of the student model in downstream tasks with fixed computation graphs. When directly adapting existing models to dynamic token pruning (Kim and Cho, 2021), we find that these models get poor performance when we adopt an aggressive pruning ratio. Therefore, to effectively combine model compression and dynamic token pruning to achieve a better speed-performance trade-off, we propose a novel distillation algorithm that not only transfers high-quality knowledge to the student but also helps the student to get adapted to dynamic token pruning, especially to a high-pruning extent. We first introduce what form of knowledge we prepare to transfer and how to transfer, then we present how to effectively get the student adapted to dynamic token pruning.

### 3.1.1 Knowledge Transfer

One of the vital problems in knowledge distillation is knowledge transfer. In transformer distillation, hidden representations (Jiao et al., 2020b), attention dependencies (Wang et al., 2020b), relations among representations (Park et al., 2021) all have been considered as useful knowledge to be transferred. Drawing on existing works, we first consider the hidden representations as the fundamental knowledge that mainly guides the learning of a student. Besides, considering that a majority of dynamic token pruning approaches (Goyal et al., 2020; Kim and Cho, 2021; Kim et al., 2021) have demonstrated that the attention scores derived from attention matrices of self-attention mechanism can act as a good indicator for token importance. Therefore, to produce a customized student model for token pruning, we additionally involve attention dependencies as the knowledge to be transferred.

To elaborate on our design in knowledge transfer, we first briefly describe the calculations of the

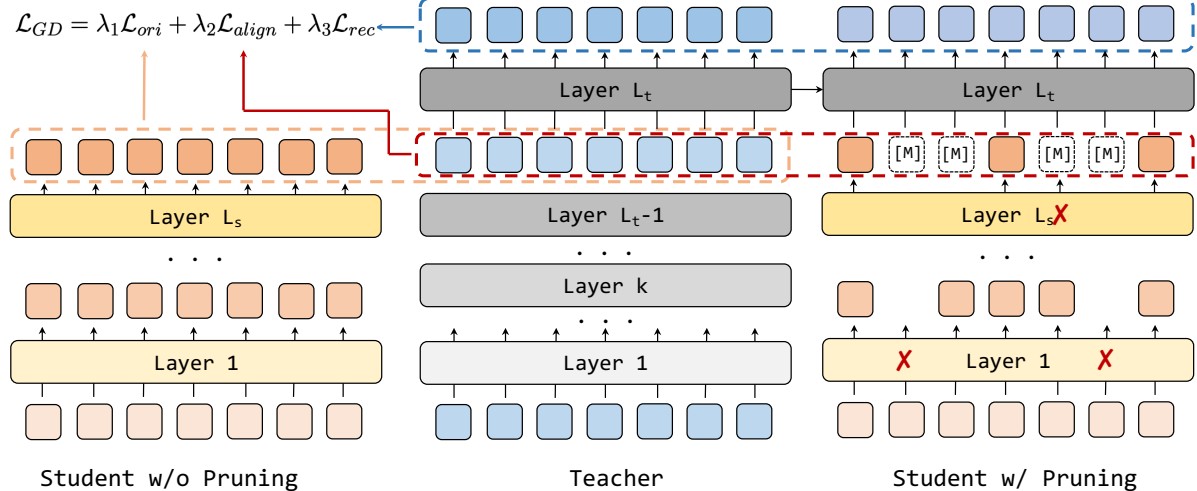

Figure 1: Illustration of general distillation in LAD. For an input text, the student has two branches of calculation: (1) the standard forward pass without token pruning on the left and (2) the forward pass with token pruning on the right. For the left part, we align the output of its last layer to the teacher's penultimate layer using $\mathcal{L}_{ori}$. For the right part, we not only align the student's remaining representations after token pruning to the corresponding representations of the teacher in the penultimate layer using $\mathcal{L}_{align}$, but also encourage the remaining representations to reconstruct the representations of the whole input with the help of the teacher's last layer as decoder using $\mathcal{L}_{rec}$.

transformer model together with some necessary notations. Formally, given a sequence of input text, a sub-word tokenizer first splits it into $n$ tokens $x = [w_1, w_2, \ldots, w_n]$. Then through an embedding layer, each token in $x$ is converted as a dense vector through a lookup table, resulting in $\boldsymbol{H}^0 = [\boldsymbol{h}_1^0, \boldsymbol{h}_2^0, \ldots, \boldsymbol{h}_n^0]$ where $\boldsymbol{h}_i^0 \in \mathbb{R}^d$. Then, the output of the embedding layer is passed to $L$ stacked transformer (Vaswani et al., 2017) layers which are mainly comprised of a multi-head self-attention module and a position-wise feed-forward network. The self-attention module produces the attention matrices $\boldsymbol{A}^l \in \mathbb{R}^{h \times n \times n}$ that encodes the dependencies among the input tokens using $h$ attention heads. The output of the $l$-th layer is denoted as $\boldsymbol{H}^l = [\boldsymbol{h}_1^l, \boldsymbol{h}_2^l, \ldots, \boldsymbol{h}_n^l]$.

With these notations, we now go back to the knowledge distillation setting. Given a teacher model with $L_t$ layers and a student model with $L_s$ layers, we feed the same text into them and can obtain the corresponding output hidden states $\{\boldsymbol{H}_t^l\}_{l=0}^{L_t}$, $\{\boldsymbol{H}_s^l\}_{l=0}^{L_s}$ and attention matrices $\{\boldsymbol{A}_t^l\}_{l=1}^{L_t}$, $\{\boldsymbol{A}_s^l\}_{l=1}^{L_s}$. We suppose the student's $l_s$-th layer is aligned with the teacher's $l_t$-th layer, then the outputs of the student (i.e., $\boldsymbol{H}_s^{l_s}$ and $\boldsymbol{A}_s^{l_s}$) should be close to the teacher's (i.e., $\boldsymbol{H}_t^{l_t}$ and $\boldsymbol{A}_t^{l_t}$). For aligning hidden states, instead of aligning sentence-level representations (Sun et al., 2020), we assume that token-level representations are more suitable for general distillation since they contain more fine-grained knowledge. Drawing inspirations from contrastive learning (Chen et al., 2020; He et al., 2020; Tian et al., 2019), we design a contrastive objective to achieve this goal. For a hidden representation of the student $\boldsymbol{h}_{s,i}^{l_s}$, we first conduct linear projection to get $\boldsymbol{z}_{s,i}^{l_s} \in \mathbb{R}^d$ that have the same hidden dimensions as the teacher model. The positive representation that it is enforced to match is the corresponding teacher's representation $\boldsymbol{h}_{t,i}^{l_t}$, while there remain multiple choices for the negatives. Different from previous works that utilize the representations of tokens in the same input of text as negatives (Tao et al., 2022), we found that simply using randomly sampled token representations performs better. As demonstrated by previous works that contrastive learning requires a large number of negatives, we employ a memory queue (He et al., 2020) $\boldsymbol{M} = \{\boldsymbol{h}_{t,j}^{l_t}\}_{j=1}^m$ with memory size $m$ to avoid the unaffordable memory cost brought by in-batch negatives with enlarged batch size (Chen et al., 2020). Then the token-level contrastive distillation objective for $x$ is formulated as:

$$\mathcal{J}_{hid} = -\sum_{i=1}^n \log \frac{\exp(s(\boldsymbol{z}_{s,i}^{l_s}, \boldsymbol{h}_{t,i}^{l_t})/\tau)}{\sum_{\boldsymbol{h}_{t,j}^{l_t} \in \boldsymbol{M}} \exp(s(\boldsymbol{z}_{s,i}^{l_s}, \boldsymbol{h}_{t,j}^{l_t})/\tau)}, \tag{1}$$

where $\tau$ is the temperature and $s(\cdot)$ denotes cosine similarity. Different from He et al. (2020) that using momentum encoder to make the stored representations stable and consistent during training,

we find that directly using the fixed teacher's representations without the help of momentum encoder yield better performance as the representations are naturally consistent. While for aligning attention dependencies, we follow Jiao et al. (2020b) to optimize the mean square error (MSE) between the attention matrices of the teacher and the student:

$$\mathcal{J}_{att} = -\text{MSE}(\boldsymbol{A}_s^{l_s}, \boldsymbol{A}_t^{l_t}). \qquad (2)$$

The overall objective for knowledge transfer is:

$$\begin{aligned} \mathcal{J}(\boldsymbol{H}_s^{l_s}, \boldsymbol{A}_s^{l_s}, \boldsymbol{H}_t^{l_t}, \boldsymbol{A}_t^{l_t}) = & \mathcal{J}_{hid}(\boldsymbol{H}_s^{l_s}, \boldsymbol{H}_t^{l_t}) \\ & + \mathcal{J}_{att}(\boldsymbol{A}_s^{l_s}, \boldsymbol{A}_t^{l_t}). \end{aligned} \qquad (3)$$

### 3.1.2 Length-Adaptive Distillation

Based on the design of knowledge transfer, we then focus on the second goal: how to effectively get the student accustomed to token pruning while transferring general knowledge. We achieve this goal by employing a teacher without token pruning to teach the student with dynamic token pruning.

**Token Pruning**   In the standard setting of dynamic token pruning, there are two major issues: how to measure the importance of tokens in each layer and how to decide the number of tokens maintained in each layer. We follow the typical solution where the importance measurement is the sum of attention scores a token received from other tokens within a sentence (Goyal et al., 2020) and the length configuration for token pruning is obtained by sampling from a pre-defined range in training and using evolutionary search (Cai et al., 2019; Wang et al., 2020a; Kim and Cho, 2021) in inference. Given an input sequence, we first sample a length configuration $\mathcal{N} = [n_1, n_2, \ldots, n_{L_s}]$ for the student with ratio $r$ with the sampling strategy proposed by Kim and Cho (2021) where $n_l$ denotes the number of tokens maintained in the $l$-th layer of the student. Hereby, the output of the student on its top layer only contains $n_{L_s}$ token representations $\hat{\boldsymbol{H}}_s^{L_s} = [\hat{\boldsymbol{h}}_{z_1}^{L_s}, \hat{\boldsymbol{h}}_{z_2}^{L_s}, \ldots, \hat{\boldsymbol{h}}_{z_{n_{L_s}}}^{L_s}]$ and the corresponding attention matrices $\hat{\boldsymbol{A}}_s^{L_s}$ formed by them, where $\mathcal{Z} = \{z_1, z_2, \ldots, z_{n_{L_s}}\}$ is the original indices of the remaining tokens. Based on $\mathcal{Z}$, we can also extract the corresponding teacher's hidden states in the teacher's $l_t$-th layer $\hat{\boldsymbol{H}}_t^{l_t}$ from $\boldsymbol{H}_t^{l_t}$ and then calculate the attention matrices $\hat{\boldsymbol{A}}_t^{l_t}$ with them.

**Training**   Based on the pruned student's representations, we propose two objectives that aim to get the student accustomed to token pruning. The first one encourages the remaining representations of the student to recover the representations of the whole input. To achieve this goal, we first construct the masked input which is the concatenation of two parts: (1) the remaining representations in the student's top layer after pruning, and (2) the query of the pruned representations constructed by adding the positional embedding of the pruned positions and the embedding of the mask token [M] token. The masked input is linearly projected to fit the hidden dimensions of the teacher. Then, we need to reconstruct the representations based on the masked input through a decoder. Instead of initializing a new decoder to predict the pruned tokens (Liu and Shao, 2022), we propose to take advantage of the teacher model by borrowing its last transformer layer as the decoder. This design not only enjoys the good ability of representation learning of the pre-trained teacher but also reduces the number of parameters that need to be optimized. The borrowed decoder is frozen during the training of the student. We denote the reconstructed outputs as $\tilde{\boldsymbol{H}}_s$ and $\tilde{\boldsymbol{A}}_s$, and form the reconstruction loss as:

$$\mathcal{L}_{rec} = \mathcal{J}(\tilde{\boldsymbol{H}}_s, \tilde{\boldsymbol{A}}_s, \boldsymbol{H}_t^{L_t}, \boldsymbol{A}_t^{L_t}) \qquad (4)$$

The second one is to align the remaining representations of the student with the corresponding ones of the teacher. Notice that in Eq. 3.1.2 we feed the representations of the $L_s$-th layer of the student to the last layer of the teacher, implicitly aligning the student's $L_s$-th layer with the $L_{t-1}$-th layer. Therefore, we form the alignment loss as:

$$\mathcal{L}_{align} = \mathcal{J}(\hat{\boldsymbol{H}}_s^{L_s}, \hat{\boldsymbol{A}}_s^{L_s}, \hat{\boldsymbol{H}}_t^{L_t-1}, \hat{\boldsymbol{A}}_t^{L_t-1}). \qquad (5)$$

In addition to the two objectives that aim to get the student accustomed to token pruning, we also transfer the knowledge without token pruning to stabilize training:

$$\mathcal{L}_{ori} = \mathcal{J}(\boldsymbol{H}_s^{L_s}, \boldsymbol{A}_s^{L_s}, \boldsymbol{H}_t^{L_t-1}, \boldsymbol{A}_t^{L_t-1}). \qquad (6)$$

The overall objective for general distillation is the weighted sum of the three objectives:

$$\mathcal{L}_{GD} = \lambda_1 \mathcal{L}_{ori} + \lambda_2 \mathcal{L}_{align} + \lambda_3 \mathcal{L}_{rec}. \qquad (7)$$

We name the model produced by general distillation as $\text{LAD}_{GD}$.

### 3.2 Task-Specific Distillation

Based on the model $\text{LAD}_{GD}$ produced by general distillation, we inject task-specific knowledge with

the task-specific teacher, while further improving the customization for dynamic token pruning under the prediction mode of the task (e.g., classification). This distillation stage is on the training set of a downstream task. As the labeled task data is usually far less than general corpora, we follow Jiao et al. (2020b); Liu et al. (2022b) to conduct data augmentation (DA) to enlarge the task data. We focus on classification tasks in this paper, therefore in task-specific distillation, we focus on transferring the knowledge held in the start token (e.g.,  in RoBERTa) instead of considering all tokens. Similar to general distillation, we sample the length configuration for token pruning with the same ratio $r$ as general distillation for the student while keeping the teacher unpruned. The (projected) representations of the start token of the student and the teacher are denoted as $\hat{\boldsymbol{z}}_{s,0}^{\boldsymbol{L_s}}$ and $\boldsymbol{h}_{t,0}^{\boldsymbol{L_t}}$. We employ contrastive distillation to transfer the knowledge with the help of the memory queue:

$$\mathcal{L}_{TD} = -\log \frac{\exp(s(\hat{\boldsymbol{z}}_{s,0}^{\boldsymbol{L_s}}, \boldsymbol{h}_{t,0}^{\boldsymbol{L_t}})/\tau)}{\sum_{\boldsymbol{h}_j \in \boldsymbol{M}} \exp(s(\hat{\boldsymbol{z}}_{s,0}^{\boldsymbol{L_s}}, \boldsymbol{h}_j)/\tau)}. \quad (8)$$

After distillation with Eq. 3.2, we fine-tune the model to finally adapted it to downstream tasks. We denote the model produced by task-specific distillation as $\text{LAD}_{TD}$ or $\text{LAD}_{TD}$ w/ DA depending on whether the training set of the task is augmented.

### 3.3 Configuration Search for Token Pruning

Given a small student model produced by our two-stage knowledge distillation framework, acceleration through dynamic token pruning can be achieved given a length configuration $\mathcal{N} = [n_1, n_2, \ldots, n_{L_s}]$ where $n_l$ denotes the number of remained tokens in layer $l$. Naturally, there is a trade-off between accuracy and efficiency with various choices of $\mathcal{N}$ and we need to find a series of optimal length configurations to facilitate various application scenarios. We achieve this goal by conducting evolutionary search (Cai et al., 2019) following Kim and Cho (2021). Specifically, we first initialize the candidate set of length configurations by random sampling, then iteratively construct new configurations by mutation and crossover operations and update the candidate set if some newly constructed configurations achieve better accuracy-efficiency trade-off. After searching, we obtain a set of optimal length configurations $\mathcal{S} = \{\mathcal{N}_1^*, \mathcal{N}_2^*, \ldots, \mathcal{N}_m^*\}$ that lies in the accuracy-efficiency Pareto frontier. In our experi-

ments, we manually select one length configuration that achieves good accuracy as well as considerable speed-up from $\mathcal{S}$ and report the results.

## 4 Experiment

### 4.1 Datasets and Metrics

We conduct experiments on 8 generally adopted tasks from GLUE benchmark (Wang et al., 2018) following previous work (Jiao et al., 2020b; Wang et al., 2021), including 2 single sentence tasks:SST-2 (Socher et al., 2013), CoLA (Warstadt et al., 2019), and 6 text pair tasks: MNLI (Williams et al., 2018), QNLI (Rajpurkar et al., 2016), RTE (Bentivogli et al., 2009), MRPC (Dolan and Brockett, 2005), STS-B (Cer et al., 2017), QQP (Chen et al., 2018). As for performance metrics, we report Spearman's rank correlation coefficient (Spear) on STS-B, Matthews correlation coefficient (Mcc) on CoLA, and accuracy (Acc) on the other 5 tasks. To evaluate the inference efficiency, we use FLOPs as the metric and employ torchprofile [2] as the tool to calculate following Kim and Cho (2021).

### 4.2 Implementation Details

We adopt RoBERTa-base (Liu et al., 2019) as the teacher model and utilize a small transformer model with 6 layers, 384 hidden dimensions and 12 attention heads as the student following Wang et al. (2021). We conduct two-stage knowledge distillation. In general distillation, we prepare English Wikipedia and BookCorpus (Zhu et al., 2015) as the training corpora and set the max sequence length as 128. We set the size of memory queue $m = 16384$, the temperature $\tau = 0.07$, the weights for different loss terms $\lambda_1 = 1.0$, $\lambda_2 = \lambda_3 = 0.5$. We randomly sample the token pruning ratio $r$ from $\{0.1, 0.2, \ldots, 0.7\}$ for each training instance for length-adaptive distillation. We use AdamW (Loshchilov and Hutter, 2017) as the optimizer and train the student model with a batch size of 256, the learning rate as 6e-4, the maximum training steps as 400k, and the warmup ratio as 0.01. In task-specific distillation, we prepare two types of datasets: the original training set of each task, and their augmentation version where each sample is augmented using contextual world replacement with the same data augmentation setting of Jiao et al. (2020b). The hyperparameters $m$, $\tau$ and $r$ are the same as general distillation. When using the original training sets, we train the student with a batch size of 32, the learning rate

[2] https://github.com/zhijian-liu/torchprofile.

| Model | SST-2 | | MRPC | | RTE | | STS-B | | MNLI-m | | QNLI | | QQP | | CoLA | |
|---|---|---|---|---|---|---|---|---|---|---|---|---|---|---|---|---|
| | Acc | S↑ | Acc | S↑ | Acc | S↑ | Spear | S↑ | Acc | S↑ | Acc | S↑ | Acc | S↑ | Mcc | S↑ |
| Pre-trained Language Models without Token Pruning | | | | | | | | | | | | | | | | |
| BERT$_{base}$ | 92.8 | 0.2× | 90.3 | 0.2× | 61.0 | 0.2× | 88.4 | 0.2× | 84.6 | 0.2× | 91.3 | 0.2× | 91.2 | 0.2× | 56.8 | 0.2× |
| RoBERTa$_{base}$ | 94.8 | 0.2× | 90.2 | 0.2× | 78.7 | 0.2× | 91.2 | 0.2× | 87.6 | 0.2× | 92.8 | 0.2× | 91.9 | 0.2× | 63.6 | 0.2× |
| General Distillation Models without Token Pruning | | | | | | | | | | | | | | | | |
| TinyBERT | 88.7 | 1.0× | 87.3 | 1.0× | 67.2 | 1.0× | 88.1 | 1.0× | 81.0 | 1.0× | 89.1 | 1.0× | 90.0 | 1.0× | 36.5 | 1.0× |
| MiniLMv2 | 90.9 | 1.0× | 86.8 | 1.0× | 66.1 | 1.0× | 88.1 | 1.0× | 81.9 | 1.0× | 89.6 | 1.0× | 90.3 | 1.0× | 40.7 | 1.0× |
| LAD$_{GD}$ | 91.6 | 1.0× | 88.7 | 1.0× | 67.9 | 1.0× | 88.5 | 1.0× | 82.3 | 1.0× | 90.2 | 1.0× | 90.7 | 1.0× | 41.0 | 1.0× |
| LAD$_{TD}$ | 93.7 | 1.0× | 89.2 | 1.0× | 68.6 | 1.0× | 88.2 | 1.0× | 84.8 | 1.0× | 91.1 | 1.0× | 91.7 | 1.0× | 44.7 | 1.0× |
| LAD$_{TD}$ w/ DA | 93.5 | 1.0× | 90.2 | 1.0× | 69.7 | 1.0× | 89.4 | 1.0× | 86.2 | 1.0× | 91.5 | 1.0× | 91.7 | 1.0× | 50.9 | 1.0× |
| General Distillation Models with Token Pruning | | | | | | | | | | | | | | | | |
| TinyBERT | 88.5 | 5.4× | 84.7 | 3.3× | 52.3 | 6.4× | 87.3 | 3.6× | 80.5 | 2.6× | 88.0 | 2.8× | 89.7 | 3.3× | 35.1 | 8.9× |
| MiniLMv2 | 89.5 | 5.7× | 84.5 | 3.4× | 57.7 | 6.2× | 87.6 | 3.6× | 81.0 | 2.7× | 88.3 | 2.8× | 90.2 | 3.5× | 40.3 | 8.6× |
| LAD$_{GD}$ | 90.4 | 5.8× | 87.5 | 3.5× | 63.5 | 7.8× | 88.1 | 3.7× | 81.5 | 2.8× | 89.5 | 2.8× | 90.5 | 3.8× | 40.8 | 8.7× |
| Task-specific Distillation / Adaptation Models with Token Pruning | | | | | | | | | | | | | | | | |
| TinyBERT w/ Ada | 88.6 | 6.2× | 85.1 | 4.0× | 60.6 | 6.3× | 87.4 | 3.2× | 80.0 | 2.7× | 88.1 | 3.2× | 89.7 | 4.0× | 36.7 | 8.4× |
| MiniLMv2 w/ Ada | 91.2 | 6.3× | 85.3 | 4.1× | 59.6 | 6.3× | 87.6 | 3.7× | 81.2 | 2.8× | 88.1 | 2.9× | 90.2 | 3.7× | 39.1 | 8.2× |
| LAD$_{TD}$ | 93.0 | 6.1× | 87.0 | 4.1× | 65.3 | 5.8× | 87.9 | 4.0× | 84.2 | 4.1× | 90.2 | 3.9× | 91.4 | 5.9× | 44.8 | 8.9× |
| LAD$_{TD}$ w/ DA | 93.3 | 7.2× | 89.9 | 4.9× | 67.1 | 8.0× | 89.1 | 4.9× | 85.6 | 4.5× | 90.2 | 4.3× | 91.4 | 6.1× | 47.6 | 8.9× |

Table 1: Performance and speedup (S↑) on GLUE benchmark. LAD$_{GD}$ denotes the model produced by general distillation, LAD$_{TD}$ denotes the model produced by general and task-specific distillation, DA denotes data augmentation for the task dataset, and Ada denotes the adaptive training baseline benchmarking against LAD$_{TD}$.

as 3e-5, the maximum training epochs as 50 for CoLA and 20 for other tasks, and the warmup ratio as 0.1. When on the augmented training sets, the batch size is 256, the learning rate is 1e-4, the warmup ratio is 0.06, and the maximum training epochs are the same as on the original training sets. In the following fine-tuning stage, we choose the learning rate from {1e-6, 2e-6, 3e-6} and the batch size from {16, 32}. For dynamic token pruning, we use the same configuration of evolutionary search as Kim and Cho (2021).

### 4.3 Baseline Methods

**Baselines for General Distillation** The first group of baselines is pure model compression methods using advanced knowledge distillation techniques. We implement TinyBERT (Jiao et al., 2020b) and MiniLMv2 (Wang et al., 2021), two representative knowledge distillation methods, under the same distillation setting (i.e., model size, training data, optimization hyperparameters, etc.) as ours. We compare two perspectives with these baselines: (1) the standard evaluation of model performance (i.e., fine-tuning on each task and testing without token pruning) and (2) the speed-performance trade-off under dynamic token pruning without any types of adaptation.

**Baselines for Task-Specific Distillation** The second type of baseline is the adaptation ap-

proach which adapts a given language model to dynamic token pruning. Following pipeline approaches (Guskin et al., 2021, 2022) that apply dynamic token pruning upon compressed small language models, we adopt the sandwich rule and inplace distillation (Yu and Huang, 2019) used in LAT (Kim and Cho, 2021) as the baseline (denoted as Ada) to be compared with our task-specific distillation method.

### 4.4 Overall Performance

We provide the overall evaluation results of the performance and the speedup of our method and baselines in Table 1. First, it can be observed from the second block that under standard evaluation setting (i.e., without token pruning), our model with general distillation (i.e., LAD$_{GD}$) consistently outperforms advanced general distillation methods on all tasks. With task-specific distillation and data augmentation, the performance can be further improved. Then we compare the performance of generally distilled models under the same token pruning setting in the third block. We find that LAD$_{GD}$ can still outperform baselines on both performance and inference speedup under the selected pruning configurations. Finally, we compare the performance of our task-specific distillation method with existing pipeline approaches with adaptation to token pruning in the bottom block. It can be seen that LAD$_{TD}$ not only achieves better performance

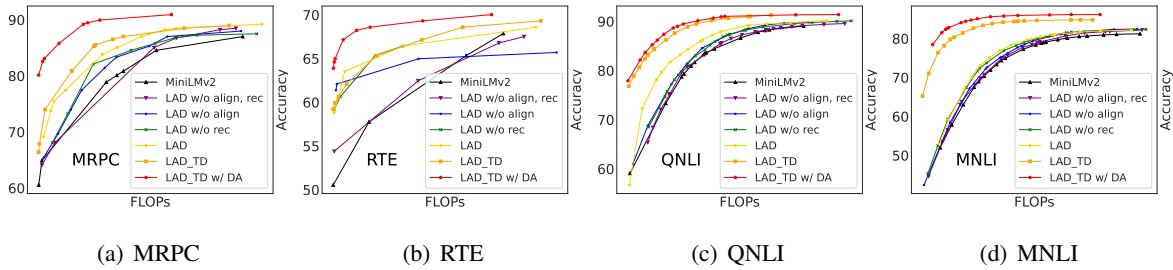

(a) MRPC      (b) RTE      (c) QNLI      (d) MNLI

Figure 2: The speed-performance curves on four tasks.

compared with $\text{LAD}_{GD}$ but also outperforms baselines. With data augmentation, we achieve dramatically better performance, where we achieve $6.1\times$ speedup while improving the performance by 2.1% on average compared with static inference without token pruning using $\text{LAD}_{GD}$.

### 4.5 Discussions

There are several critical designs in our framework. We analyze these designs by drawing speed-performance curves under different settings on four representative tasks. We choose two low-resource tasks MRPC and RTE, one moderate-resource task QNLI, and one high-resource task MNLI.

**General Distillation** In general distillation, we introduce three objectives $\mathcal{L}_{ori}$, $\mathcal{L}_{align}$ and $\mathcal{L}_{rec}$ weighted by $\lambda_1, \lambda_2, \lambda_3$ to jointly transfer general knowledge and get the student accustomed to token pruning. We first study the influence of different choices of these weights and find that setting $\lambda_1 = \lambda_2 + \lambda_3$ and $\lambda_2 = \lambda_3$ yield the best performance. Furthermore, we study the effectiveness of these objectives and plot Figure 2. Here we focus on the curves corresponding to the top 5 labels on the legend. Among these models, there are two models that are not obtained from specifically designed training for token pruning: MiniLMv2 and $\text{LAD}_{GD}$ w/o align,rec. Compared with these two baselines, we find the introduction of $\mathcal{L}_{align}$ and $\mathcal{L}_{rec}$ both improve the performance under token pruning and the combination of them performs better, verifying that our proposed length-adaptive distillation effectively help the student get accustomed to token pruning. It can also be observed that the improvement of our general distillation method over baselines under token pruning is more considerable on low-resource tasks (i.e., MRPC and RTE). The reason lies in that the fine-tuning steps on high-resource tasks are much more than on low-resource tasks, weakening the fitness for token

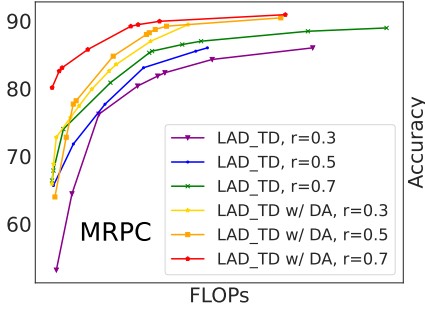

Figure 3: The speed-performance curves of different pruning ratio on MRPC.

pruning learned by our general distillation. This issue can be well addressed by our task-specific distillation algorithm discussed in the following.

**Task-Specific Distillation** Based on the generally distilled model, we further conduct task-specific distillation on the training sets of downstream tasks with or without data augmentation (DA). From Figure 2 we find that task-specific distillation brings substantial improvement for all the tasks to a different extent. For low-resource tasks (i.e., MRPC and RTE), although $\text{LAD}_{TD}$ significantly outperforms the baselines trained with no specification for token pruning (i.e., MiniLMv2 and $\text{LAD}_{GD}$ w/o align,rec), it makes a slight improvement over $\text{LAD}_{GD}$ due to limited task data. Hence with data augmentation, $\text{LAD}_{TD}$ w/ DA performs dramatically better than $\text{LAD}_{TD}$. While for tasks with moderate or abundant amounts of instances (i.e., QNLI and MNLI), we find that $\text{LAD}_{TD}$ brings considerable improvement and data augmentation is sort of the icing on the cake. These findings verify the effectiveness of task-specific distillation and prove that data augmentation is a solution to the data scarcity issue on low-resource tasks.

**The Choice of Pruning Ratio**   Recall that in both general and task-specific distillation, the student is applied with token pruning with ratio $r$. To explore the effect of the ratio as well as to find the best choice, we adopt $r \in \{0.3, 0.5, 0.7\}$ in both general and task-specific distillation on MRPC. It can be observed from Figure 3 that the larger $r$ is, the better speed-performance trade-off can be achieved whether the training set is augmented or not. More surprisingly, we also find that training with a larger pruning ratio brings consistent improvement at all speedups, which indicates that token pruning can be considered a kind of regularization in distillation that helps the student learn better.

## 5   Conclusion

In this paper, we propose a two-stage knowledge distillation framework LAD that transfers general and task-specific knowledge to the student while helping the student to get adapted to dynamic token pruning. We conduct comprehensive experiments on the GLUE benchmark. The evaluation results prove that our method can effectively take advantage of model compression and dynamic computation and achieve a superior speed-performance trade-off for inference acceleration.

## Limitations

We achieve superior speed-performance trade-off in inference acceleration by a two-stage knowledge distillation framework. In the first general distillation stage, in order to jointly transfer the general knowledge and get the student accustomed to dynamic token pruning, we introduce two calculation branches for the student. This design implies that the student needs to do two forward passes in one training iteration, increasing the computation overhead in training. In the future, we plan to explore how to unify the two computation branches to improve training efficiency.

## Acknowledgements

We would like to thank the anonymous reviewers for their constructive comments. This work was supported by the National Key Research and Development Program of China (No. 2020AAA0106600).

## Ethical Statement

In this paper, we propose a novel knowledge distillation framework that transfers knowledge to the student while customizing the student for dynamic token pruning. Our method doesn't introduce ethical issues. The datasets we used are publicly available and don't have any privacy issues.

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
