# OpenReview forum: "Length-Adaptive Distillation: Customizing Small Language Model for Dynamic Token Pruning"
_EMNLP/2023/Conference — EMNLP 2023 Findings_

### Official Review · Reviewer_YrX6 · 2023-08-02

**Soundness:** 3

**Excitement:**

3: Ambivalent: It has merits (e.g., it reports state-of-the-art results, the idea is nice), but there are key weaknesses (e.g., it describes incremental work), and it can significantly benefit from another round of revision. However, I won't object to accepting it if my co-reviewers champion it.

**Paper Topic And Main Contributions:**

The paper proposes a two-stage knowledge distillation framework LAD that transfers general and task-specific knowledge to the student while helping the student to get adapted to dynamic token pruning. The experiments on GLUE benchmark show superior performance than MiniLMv2 and TInyBERT.

**Reasons To Accept:**

Strong performance on GLUE benchmark.

**Reasons To Reject:**

1 The proposed method is a simple combination of two-stage distillation method and length pruning, which lacks of novelty.
2 The method should be compared with cascade methods (1 two-stage distillation -> pruning; 2 pruning -> two-stage distiilation) to showcase the effectiveness.

**Reproducibility:**

2: Would be hard pressed to reproduce the results. The contribution depends on data that are simply not available outside the author's institution or consortium; not enough details are provided.

**Reviewer Confidence:**

3: Pretty sure, but there's a chance I missed something. Although I have a good feel for this area in general, I did not carefully check the paper's details, e.g., the math, experimental design, or novelty.

---

> ### Author Rebuttal · Authors · 2023-08-28
>
> We sincerely appreciate your constructive comments and will make every effort to address the raised concerns.
>
> + Re Q1:
>
> As highlighted in Lines 68-72 of the paper, prior research [1, 2, 3] treated model compression and dynamic token pruning as two separate processes (i.e., applying dynamic pruning to an already compressed smaller model). In contrast, we are the first to approach model compression and token pruning as an integrated process. We propose a customized knowledge distillation algorithm referred to as LAD, which enhances the student model's robustness to token pruning throughout the distillation process. Experimental results in Table 1 (the bottom block) demonstrate that our method can achieve a superior speed-performance trade-off compared to previous pipeline approaches.
>
>
> + Re Q2:
>
> Indeed, we have compared our method with cascade methods (i.e., two-stage distillation -> pruning), and the results are presented in the bottom block of Table 1. It can be observed that our method achieves a superior speed-performance trade-off.
> As for "pruning -> two-stage distillation," we would like to emphasize that this pipeline requires compressing a model using a static pruning algorithm first and is beyond the scope of the dynamic token pruning that we studied in this paper.
>
> + Re `` Reproducibility’’:
>
> We will release our implementation to facilitate future research and provide more implementation details in appendix in our final version.
>
>
> Hope the above explanation may address your concern.
>
>
> **References**
>
> [1] Kim G, Cho K. Length-adaptive transformer: Train once with length drop, use anytime with search[J]. arXiv preprint arXiv:2010.07003, 2020.
>
> [2] Guskin S, Wasserblat M, Ding K, et al. Dynamic-TinyBERT: Boost TinyBERT's Inference Efficiency by Dynamic Sequence Length[J]. arXiv preprint arXiv:2111.09645, 2021.
>
> [3] Guskin S, Wasserblat M, Wang C, et al. QuaLA-MiniLM: a Quantized Length Adaptive MiniLM[J]. arXiv preprint arXiv:2210.17114, 2022.

---

### Official Review · Reviewer_uJ9f · 2023-08-04

**Soundness:** 3

**Excitement:**

3: Ambivalent: It has merits (e.g., it reports state-of-the-art results, the idea is nice), but there are key weaknesses (e.g., it describes incremental work), and it can significantly benefit from another round of revision. However, I won't object to accepting it if my co-reviewers champion it.

**Paper Topic And Main Contributions:**

This paper effectively combines model compression with dynamic token-pruning by proposing a reconstruction loss and an alignment loss during the knowledge distillation from a pre-trained teacher on open domain corpora. Their approach demonstrates significantly better performance-speed trade-off than strong baselines on the GLUE benchmark.

**Questions For The Authors:**

A. How will the weighting factors $\lambda_2$ and  $\lambda_3$ of the introduced losses affect the performance of their approach?
B. How is the pruning ratio decided during the inference stage of each of the tasks in GLUE?


**Reasons To Accept:**

1. The apporach provides significant performance improvement with slightly better inference speedup on the GLUE dataset.
2. The paper is well written and easy to follow.
3. The paper provides comprehensive ablation study to prove the effectiveness of the proposed additional losses

**Reasons To Reject:**

1. The paper lacks study of how the important hyper-parameters are chosen: the weighting factors $\lambda_2$ and  $\lambda_3$ of the losses. It is unclear how these hyperparameters will affect the performance of their approach.
2. The paper lacks the clarity of the implementation details, which may harm its reproducibility. It is unclear how the pruning ratio is decided during the inference of each of task in GLUE. L499-L500 indicates that the pruning ratio is set the same as the distillation step, which means that it will be sampled from a predefined set. However, if this is the case, the inference process of the models won't be deterministic, which can be an undesired property that may introduce large variance to the performance of the model.
3. The proposed method roughly doubles the training cost, while the improvements of inference speed up over previous work is relatively small.


**Reproducibility:**

3: Could reproduce the results with some difficulty. The settings of parameters are underspecified or subjectively determined; the training/evaluation data are not widely available.

**Reviewer Confidence:**

4: Quite sure. I tried to check the important points carefully. It's unlikely, though conceivable, that I missed something that should affect my ratings.

---

> ### Author Rebuttal · Authors · 2023-08-28
>
> We sincerely appreciate your constructive comments and will make every effort to address the raised concerns.
>
> + Re Reason to reject 1 and Q1:
>
> Hyper-parameters λ2 and λ3 control the intensity of the alignment loss and the recovery loss, respectively. In our paper, we heuristically set λ2 = λ3 = 0.5 to ensure a fair comparison between the two objectives and to equate the loss terms associated with token pruning to the magnitude of the original loss (i.e., λ1 = λ2 + λ3). In fact, our preliminary experiments indicated that the model's sensitivity to λ2 and λ3 is limited; various choices between 0.5 and 1.0 yield similar performance. We will incorporate this clarification in our final version. Your suggestion is greatly appreciated.
>
> + Re Reason to reject 2 and Q2:
>
> The pruning ratio is defined by a length configuration which defines how many tokens are retained in each layer. For each task in GLUE, we adopt evolutionary search following [1] to obtain a set of optimal length configurations that lies in the Pareto frontier of accuracy-efficiency trade-off. Then we manually select one length configuration that achieves good performance as well as considerable speed-up from the set. We will provide more comprehensive  details about evolutionary search in our final version. Furthermore, in response to your concern about reproducibility, we will release our implementation.
>
> + Re reason to reject 3:
>
> As demonstrated in Table 1 (lower block), our method yields markedly superior results in both performance and speed-up ratio. For instance, on the MNLI dataset, our method exhibits considerably higher accuracy (85.6 vs. 81.2) and is substantially faster than the leading baseline (4.5x vs. 2.8x).
>
> Hope the above explanation may address your concern.
>
> **References**
>
> [1] Kim G, Cho K. Length-adaptive transformer: Train once with length drop, use anytime with search[J]. arXiv preprint arXiv:2010.07003, 2020.

---

### Official Review · Reviewer_42mY · 2023-08-05

**Soundness:** 3

**Excitement:**

3: Ambivalent: It has merits (e.g., it reports state-of-the-art results, the idea is nice), but there are key weaknesses (e.g., it describes incremental work), and it can significantly benefit from another round of revision. However, I won't object to accepting it if my co-reviewers champion it.

**Paper Topic And Main Contributions:**

The paper proposes a new framework for distilling pre-trained language model, which simultaneously considers model compression and token pruning.

**Reasons To Accept:**

The idea of simultaneously considering model compression and token pruning is something that hasn't done by previous methods.

Better performance is obtained according to the experiments, illustrating the effectiveness of the proposed method.

Ablation studies are conducted, which helps better understand the components in the proposed method.

**Reasons To Reject:**

The experiments need improvement. For instance, only RoBERTa-base is used as the teacher model, and only a 6-layer transformer model with 384 hidden dimensions are used. As a distillation method, readers might be more interested in the performance of compressing larger models. Some baseline methods compared in the paper, such as MiniLM v2 (Wang et al. 2021), also did experiments on RoBERTa-large, not to mention the fact that language models have become larger and larger in recent years.

**Reproducibility:**

4: Could mostly reproduce the results, but there may be some variation because of sample variance or minor variations in their interpretation of the protocol or method.

**Reviewer Confidence:**

3: Pretty sure, but there's a chance I missed something. Although I have a good feel for this area in general, I did not carefully check the paper's details, e.g., the math, experimental design, or novelty.

---

> ### Author Rebuttal · Authors · 2023-08-28
>
> We sincerely appreciate your constructive comments and will make every effort to address the raised concerns.
>
> Re `` The experiments’’:
>
> The focus of our experiments is to validate the effectiveness of the proposed knowledge distillation algorithm, LAD. Therefore, in alignment with numerous prior works [1, 2, 3], we utilized a base-sized model (i.e., Roberta-base) as the teacher and conducted extensive experiments on all eight datasets in the GLUE benchmark. The experimental results substantiate the effectiveness of our proposed algorithm. We also consider compressing larger language models as a promising research direction and have plans to study knowledge distillation on larger models in the future.
>
> Hope the above explanation may address your concern.
>
> **References**
>
> [1] Jiao X, Yin Y, Shang L, et al. Tinybert: Distilling bert for natural language understanding[J]. arXiv preprint arXiv:1909.10351, 2019.
>
> [2] Li L, Lin Y, Ren S, et al. Dynamic knowledge distillation for pre-trained language models[J]. arXiv preprint arXiv:2109.11295, 2021.
>
> [3] Wu S, Chen H, Quan X, et al. AD-KD: Attribution-Driven Knowledge Distillation for Language Model Compression[J]. arXiv preprint arXiv:2305.10010, 2023.

---

### Meta-Review · Area_Chair_2po5 · 2023-09-19

**Recommendation:** 3

**Metareview:**

The paper presents a novel framework for distilling pre-trained language models, introducing a simultaneous approach to model compression and token pruning. This framework efficiently combines model compression with dynamic token-pruning, employing a reconstruction loss and alignment loss during knowledge distillation from a pre-trained teacher model. Through its experimentation on the GLUE benchmark, the work exhibits superior performance over contemporary methods like MiniLMv2 and TinyBERT. Notwithstanding its achievements, reviewers have raised concerns regarding the comprehensiveness of its experiments, reproducibility due to implementation ambiguity, and the increased training costs relative to the marginal improvements in inference speed.

Pros:

Introduction of a method combining model compression and token pruning, demonstrating enhanced performance on the GLUE benchmark.

Comprehensive ablation studies providing a deeper understanding of the proposed method's components.

Well-structured paper that offers clarity and ease of reading.


Cons:

Insufficient experimentation, especially concerning the compression of larger models which are becoming more prevalent.

Absence of clarity in implementation details and potential reproducibility issues, with concerns about non-deterministic inference processes.

While the proposed method offers improvements, it comes at the cost of roughly doubling the training time with only minimal gains in inference speed.

---

### Decision · Program_Chairs · 2023-10-07

**Decision:**

Accept-Findings

**Comment:**

The paper presents a novel framework for distilling pre-trained language models, introducing a simultaneous approach to model compression and token pruning. This framework efficiently combines model compression with dynamic token-pruning, employing a reconstruction loss and alignment loss during knowledge distillation from a pre-trained teacher model. Through its experimentation on the GLUE benchmark, the work exhibits superior performance over contemporary methods like MiniLMv2 and TinyBERT. Notwithstanding its achievements, reviewers have raised concerns regarding the comprehensiveness of its experiments, reproducibility due to implementation ambiguity, and the increased training costs relative to the marginal improvements in inference speed.

Pros:

Introduction of a method combining model compression and token pruning, demonstrating enhanced performance on the GLUE benchmark.

Comprehensive ablation studies providing a deeper understanding of the proposed method's components.

Well-structured paper that offers clarity and ease of reading.


Cons:

Insufficient experimentation, especially concerning the compression of larger models which are becoming more prevalent.

Absence of clarity in implementation details and potential reproducibility issues, with concerns about non-deterministic inference processes.

While the proposed method offers improvements, it comes at the cost of roughly doubling the training time with only minimal gains in inference speed.